# VLA-OS: Structuring and Dissecting Planning Representations and Paradigms in Vision-Language-Action Models

**Chongkai Gao**[1]    **Zixuan Liu**[1]    **Zhenghao Chi**[1]    **Junshan Huang**[2]    **Xin Fei**[3]

**Yiwen Hou**[1]    **Yuxuan Zhang**[1]    **Yudi Lin**[1]    **Zhirui Fang**[3]    **Zeyu Jiang**[4]

**Lin Shao**[1]

[1]National University of Singapore  [2]University of Science and Technology of China
[3]Tsinghua University  [4]Nanyang Technological University

## Abstract

Recent studies on Vision-Language-Action (VLA) models have shifted from the end-to-end action-generation paradigm toward a pipeline involving task planning followed by action generation, demonstrating improved performance on various complex, long-horizon manipulation tasks. However, existing approaches vary significantly in terms of network architectures, planning paradigms, representations, and training data sources, making it challenging for researchers to identify the precise sources of performance gains and components to be further improved. To systematically investigate the impacts of different planning paradigms and representations isolating from network architectures and training data, in this paper, we introduce VLA-OS, a unified VLA architecture series capable of various task planning paradigms, and design a comprehensive suite of controlled experiments across diverse object categories (rigid and deformable), visual modalities (2D and 3D), environments (simulation and real-world), and end-effectors (grippers and dexterous hands). Our results demonstrate that: 1) visually grounded planning representations are generally better than language planning representations; 2) the Hierarchical-VLA paradigm generally achieves superior or comparable performance than other paradigms on task performance, pretraining, generalization ability, scalability, and continual learning ability, albeit at the cost of slower training and inference speeds. Video results are in https://nus-lins-lab.github.io/vlaos/.

## 1   Introduction

Building intelligent and generalizable robots capable of perceiving, reasoning about, and interacting with physical environments remains a persistent challenge in the robotics community [34, 23]. Recent studies have increasingly emphasized the development of foundational models for robot manipulation tasks by training large Vision-Language-Action models (VLAs) on extensive datasets [8, 82, 43, 54, 2, 12, 7, 22]. Different from end-to-end foundation models in computer vision [58, 45, 40] and natural language processing tasks [1, 30, 89], recent studies of VLAs have shifted toward a new paradigm capable of performing task planning and policy learning either simultaneously or sequentially [98, 95, 27, 72, 48, 5, 77, 82]. This shift arises from the inherent complexity of robotic manipulation tasks, which naturally exhibit hierarchical structures involving both high-level task planning and low-level physical interactions [9]. Compared to end-to-end VLAs that only generate

39th Conference on Neural Information Processing Systems (NeurIPS 2025).

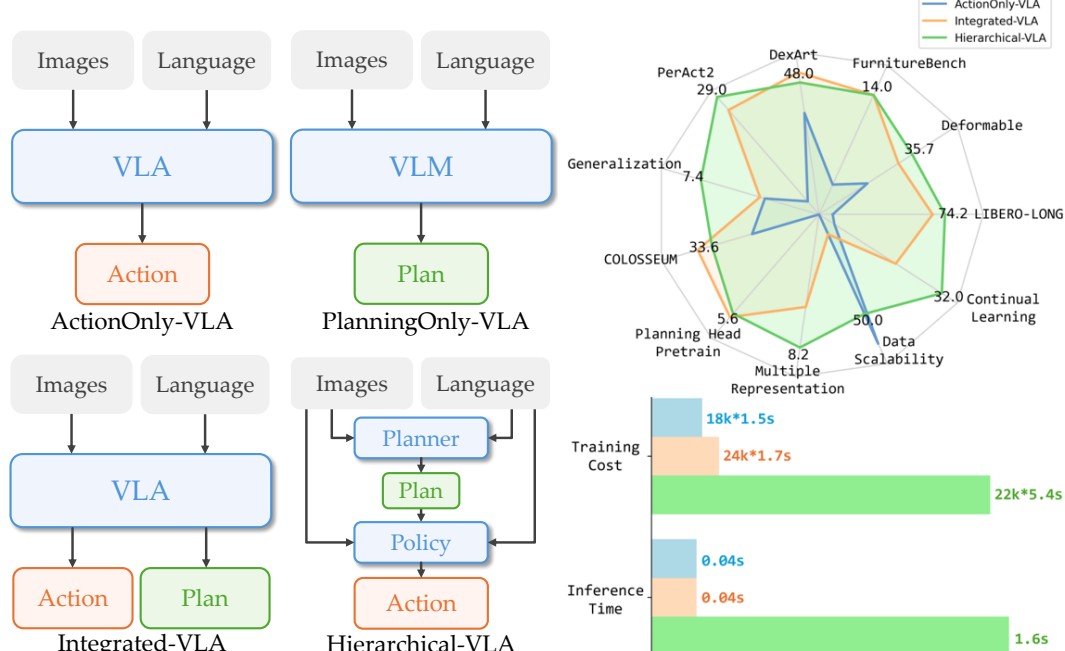

Figure 1: Left: four different VLA paradigms. Note that in this paper, we didn't explore PlanningOnly-VLA since they usually cannot be trained with the provided datasets and perform worse than others. Right: VLA paradigm comparison results. Hierarchical-VLA exhibits a generally better performance than ActionOnly-VLA and Integrated-VLA, while it incurs larger training and inference costs. This motivates future work on improving training and inference algorithms for Hierarchical-VLA models.

actions, these methods demonstrate stronger capabilities in task reasoning and comprehension for long-horizon tasks [104, 82], better success rates [95, 72], and higher sample efficiency [39, 27, 87].

However, current task-planning approaches in VLA are mainly based on intuitive designs and lack fair and systematic comparisons, as these methods vary along multiple dimensions, including network architectures, planning paradigms, data representations, and training data sources. For example, some works [72, 5, 27, 77] use a separate high-level task planning model to generate various task planning representations for a low-level VLA model, while others [82, 95, 98] use a single VLA to generate task planning representations and actions together. Consequently, substantial disagreement remains within the VLA community regarding the appropriate design and effective utilization of task planning. This makes it difficult for researchers to clearly identify which specific component contributes to performance gains or requires further improvement, hindering progress in the field.

Among these challenges, five core questions stand out: 1) **Representation**: What representation should we adopt for task planning and policy learning? Does using multiple representations yield better results, or could they conflict with one another? 2) **Paradigm**: Should we employ a monolithic model that jointly performs task planning and policy learning, or should we opt for a hierarchical paradigm where two separate models handle these tasks independently? 3) **Bottleneck**: Between task planning and policy learning, which presents a greater challenge for current manipulation tasks? 4) **Scalability and Pretraining**: Do VLAs that incorporate task planning preserve the advantageous properties of end-to-end foundation models, such as model and data scalability, as well as benefits derived from pretraining? and 5) **Performance**: Do VLAs employing task planning have better generalization and continual learning ability than end-to-end VLAs? Addressing these questions will provide the community with a clearer understanding of how task planning works in VLA models, and offer empirical evidence and guidance for future developments.

In this work, we aim to answer these questions with systematic and controllable experiments. First, to avoid biases introduced by specific neural network choices, we develop **VLA-OS**[1] model series:

---

[1]"OS" stands for "Operating System" and designates that our model family provides unified and organized interfaces of advanced VLA architectures with various planning heads and different paradigms for users.

a unified and composable family of VLA models for general-purpose manipulation tasks capable of different task planning paradigms. Concretely, we designed VLA-OS-A, VLA-OS-I, and VLA-OS-H that correspond to three mainstream VLA paradigms (ActionOnly-VLA, Integrated-VLA, Hierarchical-VLA), respectively, as illustrated in Figure 1. VLA-OS series features a unified, interchangeable VLM backbone that can be directly downloaded from HuggingFace, various plug-and-play planning heads for different representations, and two different action heads both supporting 2D/3D tasks, as shown in Figure 2. We show in Table 1 that VLA-OS exhibits superior performance compared to most existing VLA methods with fewer parameters and without pretraining.

Next, to answer the **representation** question, we annotate three kinds of task reasoning representations, including language reasoning, visual reasoning, and goal images, and conducted exhaustive combinatorial experiments with Integrated-VLA and Hierarchical-VLA models on LIBERO [51] benchmark to identify representations that yield optimal performance. Subsequently, employing the optimal representations identified, we conducted performance comparisons among three VLA paradigms on six benchmarks to answer the **paradigm** question, including rigid body manipulation tasks [51], visual generalization tasks [64], complex long-horizon tasks [32], real-world deformable manipulation tasks, dexterous manipulation tasks [3], and dual-arm manipulation tasks [28]. Furthermore, to answer the **bottleneck** question, we designed a novel set of evaluation metrics tailored to separately assess the performance of task planning and policy learning parts. To answer the **scalability** question, we use LIBERO [51] to test the model and data scalability as well as the effects of pretraining among different paradigms. And lastly, we test the generalization capabilities and continual learning ability of different VLA paradigms to answer the **performance** question.

Our experiments yield three primary findings: 1) Visually grounded planning representations (visual reasoning and image foresight planning) outperform language-based planning representations across multiple dimensions including task performance, generalization, training and speed, and low-level policy execution; 2) Hierarchical-VLA matches or exceeds the performance of Integrated-VLA and ActionOnly-VLA in terms of task performance, generalization, scalability, planning scores, continual learning, and gains from task-planning pretraining, albeit at the expense of increased training cost and slower inference; 3) On LIBERO [51] benchmark tasks, policy learning is consistently more challenging than task planning, regardless of which planning representation is used. We believe that our findings (as well as source codes, annotated datasets, and checkpoints) will provide significant help and guidance for future research within the VLA community and the broader robotics community.

## 2 Related Works

### 2.1 VLA Paradigms for Robot Manipulation

Vision-Language-Action Models (VLAs) refer to multi-modal comprehensive models that can handle visual and language inputs and generate robot actions for control. The word "VLA" was first proposed and studied in RT-2 [11], where they train a VLM to output actions as text tokens for robot control. After that, more VLA works are emerging. According to how they incorporate the task planning process, we divide VLAs into four paradigms and introduce each of them as follows.

**PlanningOnly-VLA** These works leverage pretrained LLMs or VLMs to reason and perform task planning without generating the low-level action. They break up the given task into simpler sub-tasks that can be performed by either using a set of pre-trained sub-skills [36, 2, 65, 71, 19], or outputting the parameters of pre-defined motions or cost functions for optimization [49, 73, 38, 37, 78, 57, 25, 26]. The problem is that their VLMs and low-level skills usually cannot be trained with further datasets, which frequently places them at a disadvantage compared to other VLA paradigms capable of training on given datasets [97, 92]. Consequently, we do not include PlanningOnly-VLA in this study.

**ActionOnly-VLA** These works employ an end-to-end fashion to directly map visual and language inputs to robot actions with a multi-modal network. Pioneering works mainly focus on verifying the effectiveness of large-scale robot learning [10, 11, 59, 75], while later works start to explore different model architectures, training objectives, and extra multi-modal representations and information fusion designs to make this paradigm more effective and efficient [8, 54, 43, 84, 46, 102, 62, 99, 100, 4, 103, 66]. In this work, we design VLA-OS-A for this paradigm by synthesizing several advanced model designs that have been verified to be superior in recent works [47, 8, 4].

**Integrated-VLA** These works use a single model to perform task planning and policy learning simultaneously. According to whether the action generation process is conditioned on the planning

embeddings or results, they can be further divided into explicit planning and implicit planning. For explicit planning, EmbodiedCoT [95] and CotVLA [98] generate either language-based or goal-image-based embodied chain-of-thought [79] reasoning before generating actions, and the action generation process is conditioned on the embeddings of CoT. For implicit planning, MDT [69] and PIDM [77] use goal image foresight generation loss as an auxiliary objective for planning, while RoboBrain [39] and ChatVLA [104] train VLA with auxiliary task reasoning loss in language representations. Some recent works also seek to use latent action tokens [93, 70, 13, 16, 35] that serve as forward dynamics representations to generate future images as image foresight planning, and decode these latent actions to real actions with another action head. The inputs to the action head are from the VLM encoder, and they do not need the planning heads (decoder) during inference [93, 70, 13, 16] or they only need one planning forward pass [35], so we also see these methods as implicit planning. In this work, we design VLA-OS-I for this paradigm with various plug-and-play planning heads upon VLA-OS-A for different planning representations, and design corresponding variants for both explicit and implicit planning paradigms as VLA-OS-I-I and VLA-OS-I-E.

**Hierarchical-VLA**   These works use two separate models for task planning and policy learning, with no connection or gradient between them. The idea of hierarchical models has always existed in robotics research [25, 26, 87, 14, 80]. RT-H [5] is the first work of this paradigm, where they use two identical VLMs to generate languages and actions respectively. Later works [72, 83, 82] also follow this idea but use different model architectures for task planning and action generation. Other works seek to generate multi-modal planning results for policy learning, such as image flows or trajectories [29, 27, 48], future videos [20, 91], affordance [55, 56], keypose [17], and keypoints [94]. In this work, we design VLA-OS-H for this paradigm.

## 2.2   VLA Benchmarks and Evaluations

With the rapid advancement of VLA models, benchmarks and evaluation studies for VLA have also experienced significant growth. Given the complexity and multi-dimentionality of robot manipulation tasks and VLA models, different works usually focus on evaluating one or several specific dimensions of VLA. Some works focus on the VLA model designs and training algorithms, such as different model architectures and input and output spaces [47, 88]. Other works aim to build benchmark environments and tasks to evaluate different capacities of current VLA models, such as spatial and visual generalization ability [97], long-horizon task reasoning ability [92], and different training data modalities [104]. In this work, we focus on task planning paradigms for VLA and keep the model architectures the same with systematically designed controllable experiments.

# 3   VLA-OS Model Family Design

## 3.1   Preliminaries

We study imitation learning for robot manipulation tasks. Specifically, for each task $\mathcal{T}$, we assume a set of demonstrations $\mathcal{D}_{\mathcal{T}} = \{(o_i^1, a_i^1), (o_i^2, a_i^2), \cdots, (o_i^{T_i}, a_i^{T_i})\}_{i=1}^N$ and a language goal are given, where $T_i$ is the episode length, $o$ is the observation, $a$ is the robot action, and $N$ is the number of demonstrations. We use a history of multi-view images and proprioception information as observations. In this work, we set the image resolution as $224 \times 224$. For actions, we use a normalized continuous delta end-effector pose $\delta_p$ action space and gripper open/close action $\sigma$ for training. We also let the policy generate action chunks, i.e., $a_t = ([\delta_p, \sigma]^t, \cdots, [\delta_p, \sigma]^{t+L-1})$. For dexterous hands, we use the delta joint values as the action space. We train the policy with either flow matching [50, 53] loss (for multi-modal demonstration datasets) or L1 loss (for simple and uni-modal demonstration datasets) under the suggestion of previous works [44, 8, 4, 47].

## 3.2   VLA-OS-A for ActionOnly-VLA Paradigm

VLA-OS-A model series directly generates actions without task planning stages. It is also used as the base model for other paradigms. We design a block-wise causal attention VLA drawing inspiration from [8], as shown in Figure 2. First, a VLM encodes the visual and language inputs, where the vision encoder will encode input image patches and project them into language embedding space with an MLP. Then, we use a separate set of weights as an action head for the robotics-specific tokens (action and proprioception states). The action head is a transformer decoder that has the same number of layers as the LLM, and for each layer, the queries of the proprioception tokens can attend to both the

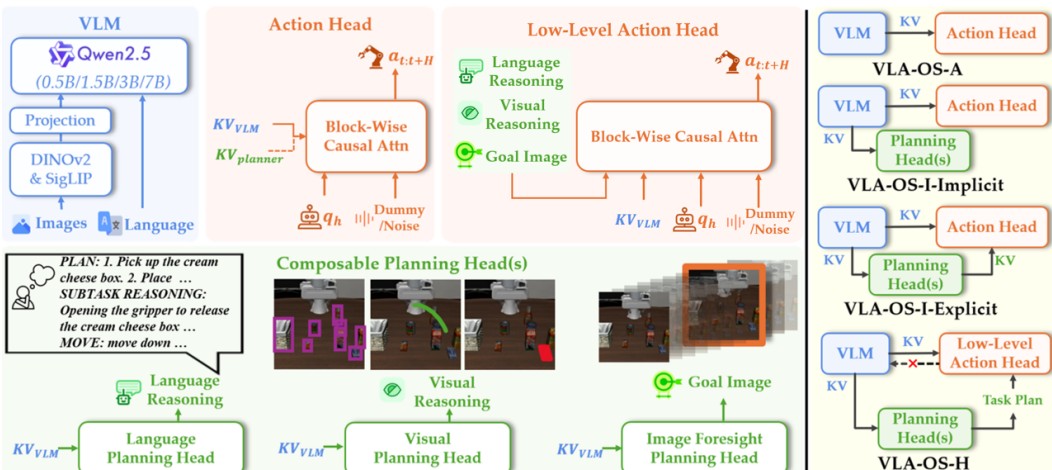

Figure 2: The VLA-OS model family. Left: the VLM and the composable heads. Our VLM has the same architecture with different numbers of parameters. Although we only draw Qwen2.5 here, our code supports any kind of LLM backbone from HuggingFace. Right: four VLA-OS architectures used in our experiments. To minimize the effects of different numbers of parameters in different models, we restrict the number of parameters of all heads to about 5% of the VLM.

keys and values from the LLM and the proprioception keys and values, and the queries of the action tokens can attend to the keys and values from the LLM, the proprioception tokens, and themselves.

Compared to $\pi_0$ [8], we make two changes in VLA-OS-A: 1) we use an ensemble of vision encoders (DINOV2 [58]+SigLIP [96]), which is proven to be better than using a single vision encoder [41]; 2) to support the model scalability experiments, we need a set of LLMs with the same structure but have different number of parameters. Thus, we choose Qwen2.5 [89] LLM series with 0.5B, 1.5B, 3B, 7B pretrained checkpoints rather than the original PaliGamma [6]. To make it a VLM, we finetune Qwen2.5 LLMs with the vision encoders and the projector on LLaVa v1.5 [52] data mixture by ourselves. We call our VLA family that uses 0.5B, 1.5B, 3B, 7B LLM backbones with suffixes of -S(mall), -B(ase), -M(iddle), and -L(arge). Detailed information can be found in Appendix C. Note, although we use Qwen2.5 in this work, our codes support any kind of LLM from HuggingFace, which makes VLA-OS highly flexible compared to [8] that is restricted to a specific backbone.

For 3D action head, we also take in multi-view depth images as input, and fuse the multi-view RGBD images to 3D point cloud using camera intrinsics and extrinsics, and inject additional CLIP features onto the point cloud, as in 3D diffuser actor [42]. We also downsample the point cloud with farthest point sampling. Each point from the downsampled point cloud will be seen as a token and these 3D tokens are sent to the action head as additional inputs.

### 3.3 VLA-OS-I for Integrated-VLA Paradigm

To perform task planning with different kinds of representations, we design three kinds of task planning heads for VLA-OS. We first annotate three kinds of task reasoning datasets corresponding to each planning representation, as shown in Figure 3. Here we only briefly introduce each of them. Details of the data annotation process can be found in Appendix B.

The **language reasoning** data contains 8 different keys [95] for each timestep, including Task, Plan, Subtask, Subtask Reason, Move, Move Reason, Gripper Position, and Object Bounding Boxes, containing the understanding of the scene and decomposition of the task. The **visual reasoning** data contains spatial semantic information and is more grounded in input images compared to language reasoning. We follow [61, 86] and use location tokens <loc i> to represent the $i$-th bin token from top-left to bottom-right. We use this kind of token to represent object bounding boxes, end-effector flow, and target object affordance as the visual planning representations. The **image foresight reasoning** data is a third-person view image at the $K$-th future step as the short-horizon goal image.

We then design language planning head, visual planning head, and image foresight planning head for each kind of representation, as shown in Figure 2. All of them are transformers that have the same number of layers with the LLM backbone, and use the block-wise causal attention mechanism to acquire the keys and values from each layer of the LLM backbone as conditions. The language

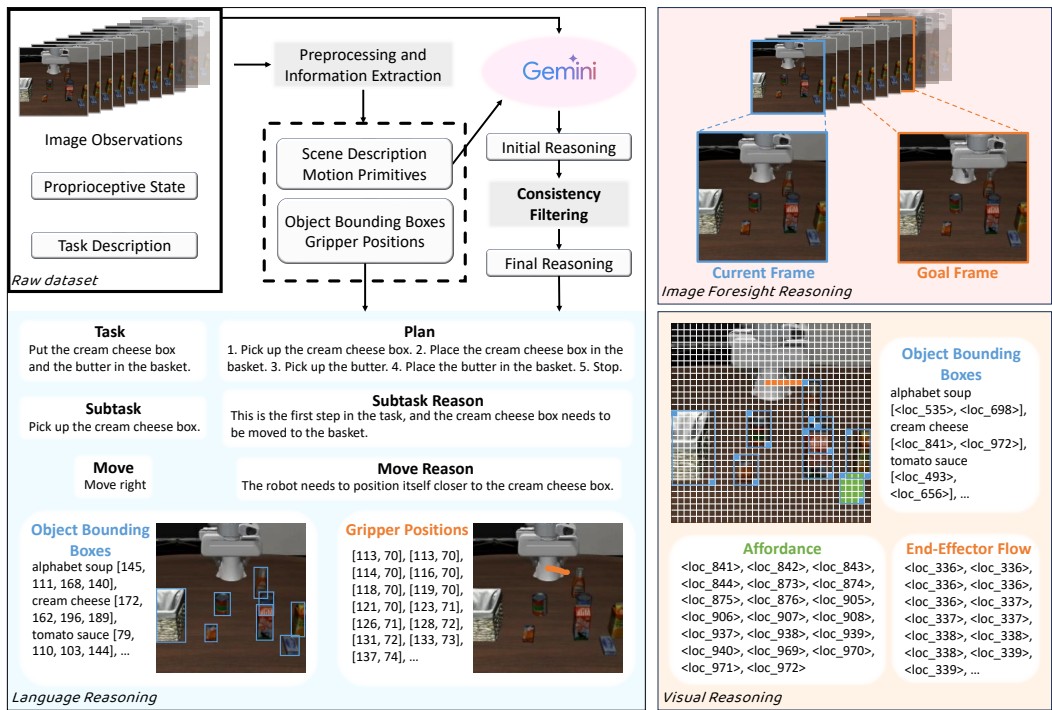

Figure 3: The formats and contents of the language reasoning dataset, the visual reasoning dataset, and the image foresight reasoning dataset in this work. We use various vision-language models for data annotation. We illustrate the language reasoning data annotation process on the top left part.

planning head uses the LLM's tokenizer for decoding, whereas the visual planning head uses an extended tokenizer vocabulary to predict location tokens. The image foresight planning head is an autoregressive image generation model similar to the recent SOTA image generator [31]. It auto-regressively generates the image in a coarse-to-fine paradigm proposed by VAR [76]. The language and visual planning heads are trained with cross-entropy loss, while the image foresight planning head is trained with the special loss in [31].

For all three planning heads, there are two kinds of ways to use them: 1) implicit planning: the action head is independent of the planning heads, i.e., the planning heads serve as auxiliary losses for the VLA training and will not be executed during inference. This may help the model avoid planning accumulation error and improve the inference speed; 2) explicit planning: the action head also attends to the keys and values from each layer of the planning heads, and during inference, the VLA must first perform task planning before generating actions. This may help solve complex tasks in a chain-of-thought [79, 95, 98] manner.

### 3.4 VLA-OS-H for Hierarchical-VLA Paradigm

This model uses two networks for task planning and policy learning respectively. As shown in Figure 2, we use the VLM together with planning heads for task planning, and modify the action head to an encoder-decoder transformer for policy learning. This action head can take as input the images, proprioception observations, and the planning representations to generate actions. To keep the comparison fair, we make the layer of the encoder and decoder of the action head half of the other two VLA-OS paradigms. We also give frozen image features from AM-Radio [67] and language features from Qwen2.5 [89] for the inputs of the action head to compensate for deficiencies in visual and linguistic features not captured by the VLM. Training details are in Appendix C.

## 4 Experiments and Findings

In this section, we perform systematic and controllable experiments with the VLA-OS model series on various manipulation tasks shown in Figure 4 to answer the research questions in Section 1. Detailed experimental settings are in Appendix C. All models are trained on 8×NVIDIA A100 80G GPUs. The continual learning experiments are in Appendix C.

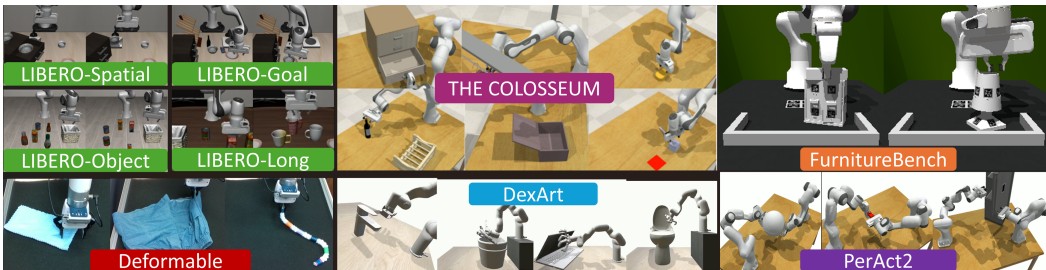

Figure 4: Benchmarks used in our evaluations, including LIBERO [51] and FurnitureBench [32] for 2D rigid body manipulation experiments, The COLOSSEUM [64] for 3D and generalization evaluation, real-world deformable object manipulation tasks (fold the handkerchief, unfold the jean, and straighten the rope), DexArt [3] for dexterous tasks, and PerAct2 [28] for dual-arm tasks.

## 4.1 Sanity Check of VLA-OS

Before investigating different VLA paradigms for our research questions, we first verify the correctness and basic performance of our VLA-OS models to serve as a foundational sanity check. We train VLA-OS-A-S on four suites from LIBERO [51] (LIBERO-Spatial, LIBERO-Object, LIBERO-Goal, LIBERO-Long) from scratch with L1 loss and compare them with Diffusion-Policy [18], fine-tuned OpenVLA [43], fine-tuned CoT-VLA [98], fine-tuned DiT Policy [33], and the state-of-the-art methods: fine-tuned $\pi_0$ [8] and its variant $\pi_0$-FAST [62]. Results are shown in Table 1.

Table 1: Sanity check. Success rates on four LIBERO benchmarks. Baseline results are from their papers [43, 8, 44]. Our results are the average of top-3 checkpoints averaged over 20 rollouts for each task suite. **Bold** indicates the best result except SOTA, and underline indicates comparable result.

| | LIBERO-Spatial | LIBERO-Object | LIBERO-Goal | LIBERO-Long | Average |
|---|---|---|---|---|---|
| Diffusion Policy [18] (scratch) | 78.3 | 92.5 | 68.3 | 50.5 | 72.4 |
| OpenVLA [43] (fine-tuned) | 84.7 | 88.4 | 79.2 | 53.7 | 76.5 |
| CoT-VLA [98] (fine-tuned) | 87.5 | 91.6 | 87.6 | **69.0** | 81.1 |
| DiT Policy [33] (fine-tuned) | 84.2 | 96.3 | 85.4 | 63.8 | 82.4 |
| $\pi_0$-FAST [62] (fine-tuned) | **96.4** | **96.8** | 88.6 | 60.2 | 85.5 |
| VLA-OS-A-S (scratch, ours) | 87.0 | 96.5 | **92.7** | 66.0 | **85.6** |
| $\pi_0$ [8] (fine-tuned, SOTA) | 96.8 | 98.8 | 95.8 | 85.2 | 94.2 |

We can see that VLA-OS-A-S performs better ($+13.2\%$) than Diffusion Policy (trained from scratch) and the fine-tuned OpenVLA model ($+9.1\%$), CoT-VLA ($+4.5\%$), and DiT Policy ($+3.2\%$), and is comparable to fine-tuned $\pi_0$-FAST ($+0.1\%$). Although our model is worse than the SOTA method, these results sufficiently demonstrate that our model design is excellent and competitive. Note that VLA-OS-A-S is **trained from scratch** and utilizes **only a 0.5B LLM backbone**.

*Finding 1*: *For downstream tasks, larger VLA models trained on large-scale datasets do not necessarily outperform smaller models that are trained from scratch. Model architectures and algorithmic designs are still important at the current moment.*

## 4.2 Planning Representation Experiments

To explore which representation is better for task planning and policy learning, we perform comprehensive experiments with language planning (**L**), visual planning (**V**), image foresight planning (**IF**), and their combinations on LIBERO-LONG [51] benchmark that contains 10 long-horizon tasks with 50 demonstrations in each task for VLA-OS-I and VLA-OS-H. The best representation will be used as the default representation for all later experiments. Table 2 shows the results.

*Finding 2*: *For Integrated-VLA paradigm, implicit planning can yield a positive performance gain, whereas explicit planning incurs a significant performance degradation when trained from scratch.*

*Analysis*: The implicit planning paradigm leverages various auxiliary task planning objectives as additional losses for training, and during inference, there is no difference between it and ActionOnly-VLA, thus it brings performance improvement. This shows that **using task planning as auxiliary losses** can improve the performance. However, the explicit planning paradigm will have to first complete the entire planning process before the action head generation during inference, and this will bring severe

Table 2: Different planning representation comparison on LIBERO-Long. All results are the average of top-3 checkpoints averaged over 20 rollouts. Numbers in parentheses indicate the change relative to the result of VLA-OS-A in Table 1.

| | L | V | IF | L+V | L+IF | V+IF | L+V+IF |
|---|---|---|---|---|---|---|---|
| VLA-OS-I-I | 68.0 (↑2.0) | 71.0 (↑5.0) | 72.5 (↑6.5) | 66.7 (↑0.7) | **73.3** (↑7.3) | 71.0 (↑5.0) | 71.7 (↑5.7) |
| VLA-OS-I-E | 60.5 (↓5.5) | 52.5 (↓13.5) | **67.5** (↑1.5) | 42.7 (↓23.3) | 56.7 (↓9.3) | 56.7 (↓9.3) | 50.7 (↓15.3) |
| VLA-OS-H | 63.5 (↓2.5) | 69.0 (↑3.0) | 71.7 (↑5.7) | 71.5 (↑5.5) | 72.0 (↑6.0) | 73.7 (↑7.7) | **74.2** (↑8.2) |

| | VLA-OS-A | VLA-OS-I | VLA-OS-H |
|---|---|---|---|
| LIBERO-LONG (2D) | 66.0 | 73.3 (↑ 7.3) | **74.2** (↑ 8.2) |
| The COLOSSEUM (3D) | 34.4 | **35.7** (↑ 1.3) | 35.3 (↑ 0.9) |
| Deformable (Real-World) | 28.5 | **35.4** (↑ 6.9) | 33.6 (↑ 5.1) |
| FurnitureBench | 11.0 | **14.0** (↑ 3.0) | **14.0** (↑ 3.0) |
| DexArt | 45.0 | **49.0** (↑ 4.0) | 48.0 (↑ 3.0) |
| PerAct2 | 21.0 | 28.0 (↑ 7.0) | **29.0** (↑ 8.0) |
| Generalization | 6.1 | 6.2 (↑ 0.1) | **7.4** (↑ 1.3) |
| Planning Head Pretraining | – | 79.1 (↑**5.8**) | 79.8 (↑5.6) |

Training Cost for VLA-OS-H

L: 22k*5.4s
V: 24k*3.9s
IF: 15k*1.7s

Inference Time for VLA-OS-H

L: 1.92s
V: 1.76s
IF: 0.09s

(a) Success rates of different VLA paradigms on more benchmarks, as well as the generalization and task planning pretraining experiments. All results are averaged over 20 rollouts among 3 best checkpoints.

(b) Training cost and inference time for different representations.

Figure 6: More results for different paradigms and inference time and training cost for different representations. Results of the right figure are calculated from the LIBERO-LONG benchmark.

**planning accumulation error** issues. Typically, the length of planning tokens significantly exceeds that of action tokens (approximately 2000 vs. 8), which will exacerbate the accumulation error issue than purely with action tokens. Additionally, the embeddings from every layer of the planning head are fed into the action head, affecting its internal representations. Meanwhile, the action head does not receive raw visual or language inputs. It only receives embeddings from the VLM and planning heads, which makes it lack the necessary error-correction capability. Instead, Hierarchical-VLA will not only take in the raw visual observation and language instruction as inputs, but also confine the planning accumulation errors exclusively to the explicit representation level, rather than allowing them to propagate into the deeper embedding layers.

For qualitative comparisons, we show in Figure 5 an example that when VLA-OS-H uses the same planning heads as VLA-OS-I-E where there are some planning errors, it can correct the behavior while VLA-OS-I-E cannot.

*Finding 3: Visually grounded planning representations work better than language planning representations, and also have faster inference speed and smaller training cost.*

From the results in Table 2, we can see that visual planning and image foresight planning are better than language planning (↑5.75 v.s. ↑2.0 for VLA-OS-I-I and ↑4.35 v.s. ↓2.5 for VLA-OS-H). We also illustrate the inference speed and training cost in Figure 6b (introduced in Section 4.5) to show the speed and cost advantages of visually grounded planning representations.

*Finding 4: When employing multiple planning representations concurrently, Hierarchical-VLA outperforms Integrated-VLA paradigms.*

### 4.3 More Performance, Generalization, and Benefit from Planning Head Pretraining

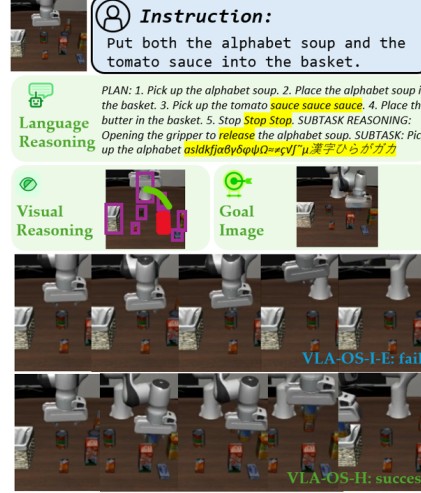

Figure 5: Comparison between VLA-OS-I-E and VLA-OS-H with the same planning errors. The three planning representations shown in this figure all have small planning errors (highlighted).

To further compare different planning paradigms, we perform additional experiments to explore their performance on: 1) more manipulation benchmarks including 3D manipulation tasks [64], real-world deformable tasks, furniture assembly tasks [32], dexterous manipulation tasks [3], and dual-arm manipulation tasks [28]; 2) generalization ability; and 3) benefits from planning head pretraining.

For 1), in COLOSSEUM, we train and test on the *No-Perturbation* setting. For 2), we use THE COLOSSEUM and train on *No-Perturbation* but test on *ALL-Perturbation* setting, including changes in color, texture, size of objects, table-tops, backgrounds, lighting, distractors, physical properties, and camera poses. For 3), a lot of literature [27, 48, 13, 72, 94, 91] claim that the primary advantage of using task planning in VLA rather than ActionOnly-VLA is that their task-planning components can be trained on large-scale task-agnostic planning data without costly action annotations. Here, we train them on LIBERO-90, a larger dataset with 90 manipulation tasks and 50 demonstrations for each task. We only train the planning components, i.e., the VLM and planning heads. Then we fine-tune the pretrained VLM and planning heads together with the action head on LIBERO-LONG with both the task reasoning and policy learning losses. Results are in Table 6a.

*Finding 5*: *Integrated-VLA and Hierarchical-VLA outperform ActionOnly-VLA across a broad spectrum of tasks (2D, 3D, simulation, and real-world), with their performances largely comparable.*

*Finding 6*: *Both Integrated-VLA and Hierarchical-VLA benefit similarly from task-planning pretraining, exhibiting analogous gains in task success rate.*

*Finding 7*: *Hierarchical-VLA demonstrates the best generalization ability.*

## 4.4 Separate Investigation of Task Planning and Policy Learning Parts

It is imperative to discern whether task failures arise from the planning component or policy learning. In this part, we use LIBERO-LONG [51] for Integrated-VLA (only for task planning) and Hierarchical-VLA to separately evaluate the task planning part and policy learning part of the model for all three planning representations. For evaluation, we manually divide each long-horizon task into several sub-tasks, and forcibly reset the environment to the initial

Table 3: Separate evaluation of task planning and policy learning modules for different paradigms and representations. Results are averaged from 20 episodes for each task in LIBERO-LONG.

|  | L | | V | | IF | |
|---|---|---|---|---|---|---|
|  | DCS | IFS | DCS | IFS | DCS | IFS |
| VLA-OS-I-I | 0.79 | – | 0.83 | – | 0.92 | – |
| VLA-OS-H | 0.81 | 0.84 | 0.86 | 0.93 | 0.94 | 0.90 |

state of each subtask. Then we compute the average planning correctness $\mathbb{I}$ (0 or 1) of the planning outcomes and execution success rate $\mathbb{S}$ (0 or 1) from the action head across all subtask start points. Thus, for a given task trajectory, we can get **Decomposition Score (DCS)**$= \frac{1}{T}\sum_{t=1}^{T}\mathbb{I}(p_t)$ and **Instruction Following Score (IFS)**$= \frac{1}{T}\sum_{t=1}^{T}\mathbb{S}(a_t^{seq})$, where $T$ is the total sub-task number and $p_t$ and $a_t^{seq}$ are the planning outcomes and actions generated at subtask $t$. Note for Hierarchical-VLA, we give the ground truth planning results when testing IFS. Results are shown in Table 3.

*Finding 8*: *Hierarchical-VLA performs better than Integrated-VLA in task planning.*

*Finding 9*: *Visually grounded planning representations are easier for low-level policy to follow.*

## 4.5 Training Cost and Inference Speed

We report the inference speed and training cost for different paradigms and planning representations. The training cost is calculated by multiplying the total training steps by the per-step time on LIBERO-LONG with $8\times$ A100 NVIDIA GPUs. Results are shown in Figure 1 and 6b.

*Finding 10*: *The autoregressive property of the language-planning representation head is the principal cause of its higher training cost and slower inference speeds.*

## 4.6 Data and Model Scalability Experiments

In this part, we perform experiments for data and model scalability of different VLA paradigms. For data scalability, we use LIBERO-LONG [51], a dataset with 10 tasks with a total of 500 demonstrations. We use 10%, 40%, 70%, and 100% of the data to train on three VLA paradigms with the model size S. For model scalability, we use LIBERO-90, a dataset with 90 tasks and 4,500 demonstrations, for the experiment with all training data. We choose Qwen-2.5 LLM backbone with parameters of 0.5B, 1.5B, 3B, and 7B for experiments. Results are shown in Figure 7.

*Finding 11*: *The performance of all VLA paradigms improves as the amount of action-labeled demonstration data increases, i.e., all VLA paradigms have the data scalability.*

*Finding 12*: *For tasks trained from scratch with roughly 5,000 demonstrations, the LLM backbone should be limited to 0.5B parameters, or keep the total model size under 1B parameters.*

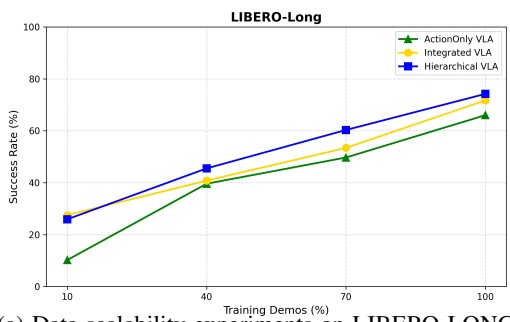
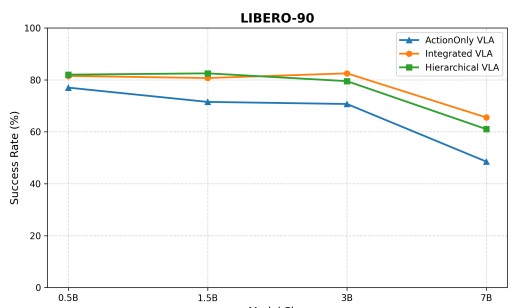

(a) Data scalability experiments on LIBERO-LONG with different planning paradigms with 0.5B LLM backbone. Success rates are calculated among 20 evaluation episodes among the 3 best checkpoints.

(b) Model scalability experiments on LIBERO-90 of all VLA paradigms. Success rates are calculated among 20 evaluation episodes among the 3 best checkpoints. We select the best checkpoint before 100k steps.

Figure 7: Data and model scalability experiments across different VLA paradigms.

## 5 Conclusion and Limitation

We provide a systematic investigation across different VLA paradigms and task planning representations through various kinds of manipulation tasks. Experiments show the superiority of visually grounded planning representations and the Hierarchical-VLA paradigm. Main findings include:

1. The time has not yet come to scale up VLA model sizes. Model architectures and training algorithms still matter.

2. Visually grounded representations (visual and image foresight) are better than language representations in terms of success rates, low-level following, and continual learning.

3. Integrated-VLA and Hierarchical-VLA outperform ActionOnly-VLA on task performance and generalization ability, but incur faster forgetting.

4. Integrated-VLA and Hierarchical-VLA perform comparably on task performance and Planning Head Pretraining, but Hierarchical-VLA generalizes better, has better task-planning performance, and performs better when using multiple planning representations.

5. All VLA paradigms have data scalability. For tasks trained from scratch with roughly 5,000 demonstrations, the LLM backbone should be limited to 0.5B parameters, or keep the total model size under 1B parameters.

We believe our findings offer meaningful insights that can inform future research in VLA and the broader robotics community. We recommend following research directions based on our findings:

1. Why are visually grounded representations better than language? The point is, language representations already contain enough information for doing the task.

2. In both explicit v.s. implicit and Hierarchical v.s. Integrated comparisons, reducing the influence (or coupling) of action head training on VLM improve the performance. We suppose this is because of the gradient conflicts between action training and planning training. So, how to verify this, why they conflict with each other, and how to avoid the gradient conflict between them are interesting questions.

3. How to design network architectures to extract information from VLM effectively? The current KV cache mechanism is good, but it restricts the layer of the planning and action heads to match the layer number of the LLM backbone.

4. How to design faster planning heads for autoregressive planning heads?

5. How to design better low-level action heads with better planning-following ability?

6. How to construct large-scale task planning datasets for VLA? How to transfer current datasets to task planning datasets?

The limitations of this paper are: 1) despite the VLA-OS family encompassing a wide array of task planning paradigms for VLA, there remain several designs and variants that we have not yet covered, such as using latent actions [93, 13] for image generation rather than VAR [76, 31]-like generator in VLA-OS, video generation for planning [91, 20], and scene flow for planning [27, 81]; 2) we didn't explore embodiment transfer, sim2real transfer, and 2D to 3D transfer problems for VLA; 3) our training dataset remains limited to fewer than 10,000 trajectories, and we have not yet investigated the research questions that arise from pretraining on larger datasets such as the OXE [60] dataset.

# 6 Acknowledgment

We thank Zhixuan Xu for his valuable discussion and his guidance for drawing the pictures.

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

# A    Benchmarks and Dataset Details

## A.1    VLM Pretraining Dataset

The LLM backbones we choose are Qwen-2.5 [89] series. Since they are not VLM, we first pretrain it to VLM with LLaVa v1.5 [52] data mixture, which consists of two subsets used for a multi-stage training pipeline. The first subset consists of a 558K sample mixture of examples sourced from various captioning datasets, while the second consists of 665K multimodal instruct tuning examples comprised of synthetic data generated in [52], as well as examples from existing vision-language training sets. According to the conclusion from Prismatic-VLMs [41], we only use the first subset to train the VLM in a single-stage optimization procedure, that is, directly fine-tuning all parameters. We implement the training code with PyTorch using Fully Sharded Data Parallel (FSDP [101]) and BF16 mixed precision and train the VLM with 2 epochs for all Qwen2.5 model types (0.5B, 1.5B, 3B, and 7B). The training hyperparameters are shown in Table 4.

Table 4: Training hyperparameters of VLM for Qwen2.5 LLM.

| Hyperparameter | Value |
|---|---|
| Batch Size | 64 |
| Max Gradient Norm | 1.0 |
| Weight Decay | 0.1 |
| Learning Rate | 2e-5 |
| Optimizer | AdamW |
| Scheduler | Warmup & Cosine Decay |
| Warmup Ratio | 0.03 |

## A.2    LIBERO Dataset

The LIBERO Dataset [51] contains five subsets: LIBERO-Spatial, LIBERO-Object, LIBERO-GOAL, LIBERO-LONG, and LIBERO-90. The first four subsets contain 10 tasks for each of them, with 50 demonstrations for each task. The last subset contains 90 tasks with also 50 demonstrations for each task. All tasks have a language instruction that describes the task. We use the two camera views for all subsets (the agentview and eye-in-hand view). We use a resolution of $224 \times 224$ for each view of the image. The action space is 7-dim, containing 6-dim $\delta x, \delta y, \delta z, \delta roll, \delta pitch, \delta yaw$ and 1-dim gripper open/close. We use a history length of 2 and a future action length of 8.

Following OpenVLA [43], we further clean up the original LIBERO datasets by:

- We filter out all "no-op" actions from the dataset, i.e., actions that have near-zero magnitude in the translation and rotation components and do not change the state of the robot's gripper.

- We replay all demonstrations in the corresponding simulation environments and filter out the demonstrations that fail to complete the task (as determined by the environments' success criteria).

## A.3    The COLOSSEUM Dataset

For 3D manipulation tasks and generalization experiments, we use The Colosseum [64] as our task benchmark. This benchmark contains 20 single-arm manipulation tasks in simulation. Each task has various variants such as lighting, distractors, physical properties perturbations, and camera pose. The cameras in this benchmark are depth cameras, so we can get the depth map and then get the point cloud observations by fusing all cameras. We follow 3D-DA [42] to preprocess the 3d observations to point cloud tokens. Then we send the point cloud tokens to the action head (or the low-level action head) as additional inputs, together with the original multi-view images. This makes the action heads have the 3D-aware property. For each task, we have 100 demonstrations. The action space is 8-dim, containing 3-dim $\delta x, \delta y, \delta z$ and 4-dim $\delta w, \delta q_x, \delta q_y, \delta q_z$ as the delta quaternion for rotation, and 1-dim gripper open/close. We use a history length of 2 and action length of 8.

## A.4   The Real-World Deformable Manipulation Dataset

For deformable object manipulation tasks, we design three real-world deformable object manipulation tasks: unfold the jeans, fold the handkerchief, and straighten the rope, as shown in Figure 4. We use two camera views for these tasks, where a third-view camera is mounted on another X-Arm, and an eye-in-hand-view camera is mounted on the main X-Arm, as shown in Figure 8. We collect 100 demonstrations for each task with human teleoperation. The cameras we use are RealSense D435i. We freeze the rotation of the X-Arm, so the action space is 4-dim: 3-dim $\delta x, \delta y, \delta z$ and 1-dim gripper open/close. The average horizon of these tasks is 214 steps. We also use an observation history length of 2 and a future action length of 8.

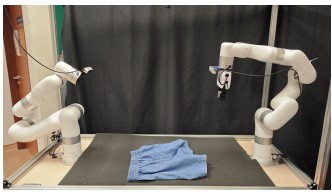 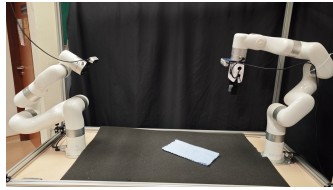 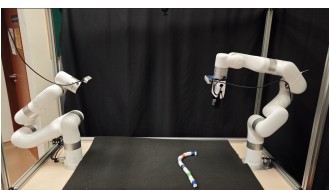

(a) The jeans unfold task.  (b) The handkerchief folding task.  (c) The rope straightening task.

Figure 8: The real-world deformable object manipulation tasks.

## A.5   The DexArt Dataset

We use the DexArt [3] benchmark for dexterous manipulation tasks. This benchmark contains four dexterous manipulation tasks built on the Sapien [85] simulator, including *turn on the faucet*, *open the laptop*, *lift the bucket*, and *open the toilet*. The original benchmark is a reinforcement learning benchmark, and they provide the official trained policy checkpoint. We load these checkpoints and collect 100 demonstrations for each task. We use one camera view for each task.

## A.6   The FurnitureBench Dataset

For long-horizon complex manipulation tasks, we choose FurnitureBench [32] as our task benchmark. This benchmark provides corresponding simulation environments called FurnitureSim, and it provides demonstrations for four tasks: *cabinet*, *lamp*, *one-leg*, and *round-table*. Each task has 100 demonstrations. The action space is 8-dim, containing 3-dim $\delta x, \delta y, \delta z$ and 4-dim $\delta w, \delta q_x, \delta q_y, \delta q_z$ as the delta quaternion for rotation, and 1-dim gripper open/close. We use three camera views as input.

## A.7   The PerAct2 Dataset

For dual-arm manipulation tasks, we choose PerAct2 [28] as our task benchmark. We use five tasks in this benchmark: *handover item*, *lift ball*, *put bottle in fridge*, *straighten rope*, and *sweep to dustpan*. As in The Colosseum, we make this benchmark a 3D task benchmark. Each task has 100 demonstrations. The action space is 22-dim, where 16-dim is for the dexterous hand joint values and 6-dim is for the end-effector. For this dataset, we do not use the image foresight planning.

## A.8   The Real-World Rigid-Body Manipulation Dataset

To further verify our conclusions in the real-world setting, we design 5 manipulation tasks in a single-arm manipulation setting, as shown in Figure 9. We collect 50 demonstrations for each task. The action space is 7-dim.

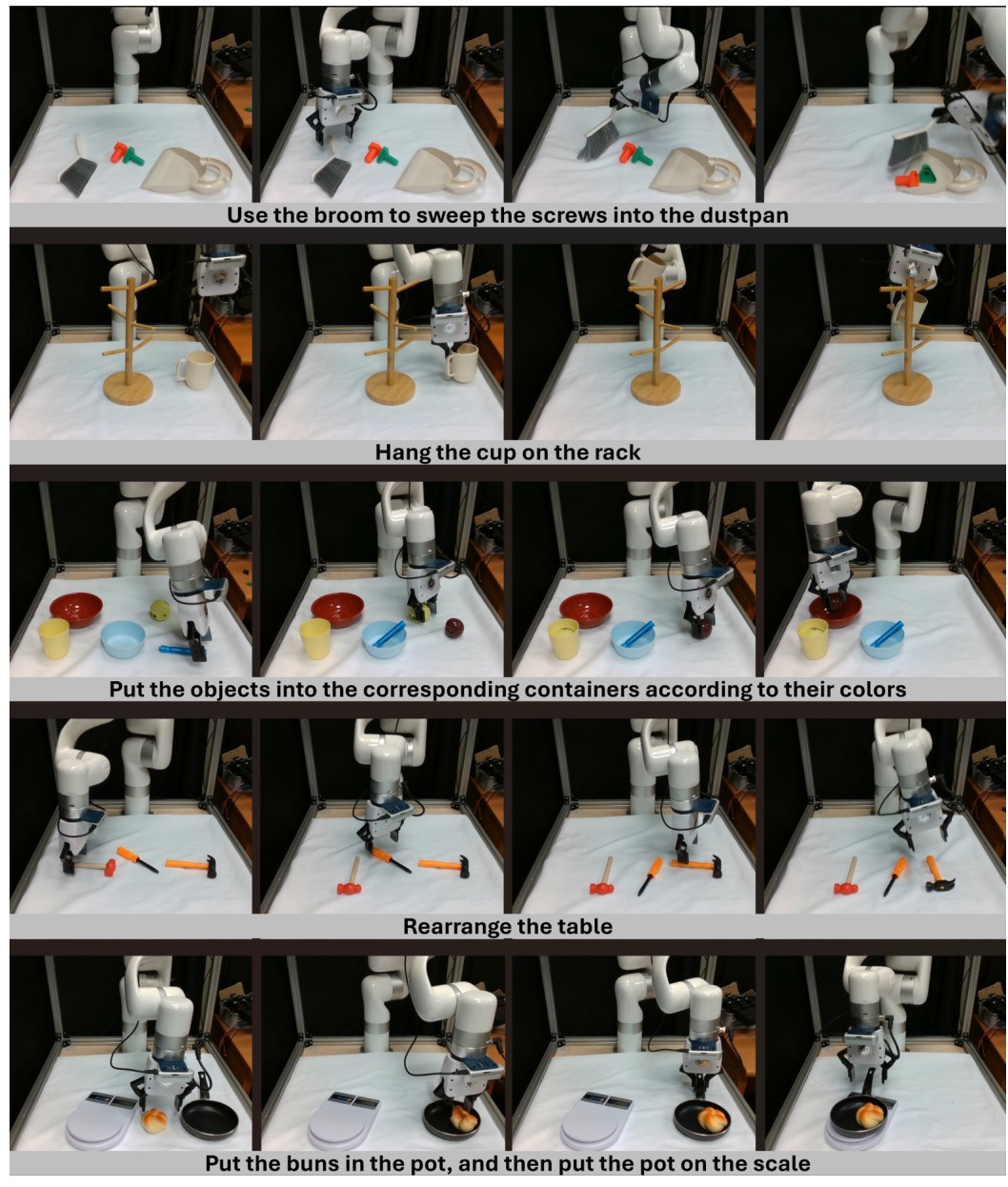

Figure 9: Real world manipulation tasks.

# B  Reasoning Dataset Annotation

## B.1  Language Reasoning Dataset

This dataset contains language-based planning results for the task that understands the scene and decomposes the task, as used in [72, 92, 104, 5]. We design a unified language planning format and structure applicable to all manipulation tasks with 8 different keys, including `Task`, `Plan`, `Subtask`, `Subtask Reason`, `Move`, `Move Reason`, `Gripper Position`, and `Object Bounding Boxes`. For example, for the task *open the top drawer of the cabinet*, the reasoning data should be:

```
TASK: Open the top drawer of the cabinet. PLAN: 1. Approach the
cabinet. 2. Locate the top drawer. 3. Locate and grasp the drawer
handle. 4. Open the drawer. 5. Stop. VISIBLE OBJECTS: akita black bowl
 [100, 129, 133, 155], plate [17, 131, 56, 158], wooden cabinet [164,
75, 224, 175] SUBTASK REASONING: The top drawer has been located; the
robot now needs to position itself to grasp the handle. SUBTASK:
Locate and grasp the drawer handle. MOVE REASONING: Moving left aligns
 the robot's end effector with the drawer handle. MOVE: move left
GRIPPER POSITION: [167, 102, 166, 102, 165, 102, 164, 102, 162, 102,
161, 102, 160, 102, 158, 102, 156, 102, 154, 102, 153, 102, 151, 102,
149, 102, 147, 102, 145, 102, 143, 102]
```

Similarly to EmbodiedCoT [95], we provide an overview of our data labeling pipeline in Figure 3. To obtain a comprehensive understanding of the scene, we first query the Prismatic-7B VLM [41], which outputs a detailed scene description. Next, we derive low-level motion primitives by analyzing the robot's proprioceptive state across a 10-step prediction horizon, assuming a static camera viewpoint, and translating these movement traces into a set of pre-defined action templates (e.g., "move left", "move up"). To construct the full reasoning trace, we use Gemini1.5 [74] to synthesize higher-level plans. Given the task instruction, scene description, and step-wise movements, Gemini1.5 generates a structured plan that includes a sequence of sub-tasks, as well as the specific sub-task relevant to each step. Additionally, it provides concise justifications for both the movement taken and the associated sub-task.

However, during experiments, we observed that the quality of the generated reasoning, referred to as *initial reasoning* in Figure 3, was often suboptimal, exhibiting two major issues. First, there was inconsistency in the planning outputs: even for the same task, the language descriptions of sub-tasks varied significantly. This stems primarily from the inherent randomness in responses from large language models such as Gemini1.5. Second, we found a mismatch between the generated plans and the actual trajectories. This issue was particularly pronounced in complex, long-horizon tasks (e.g., FurnitureBench [32]), where the provided inputs—task instruction, scene description, and step-wise movement primitives—were insufficient for the model to infer coherent and accurate planning steps. As a result, the low quality of the *initial reasoning* posed challenges for training the planning head, as the model struggled to learn meaningful mappings from observations to such plannings.

To address these issues, we applied a filtering and refinement process to the *initial reasoning*. Specifically, for each task, Gemini or human experts selected and edited the task descriptions and high-level plans produced by Gemini to ensure consistency across episodes of the same task. Once a canonical task and plan were established, we prompted Gemini again to regenerate the step-wise reasoning under this fixed structure. This process yielded the *final reasoning* in Figure 3, which aligns better with the trajectories and provides more coherent supervision for training the planning head.

In addition to the reasoning generated by Gemini, we also incorporate object bounding boxes and gripper positions into the final annotations. For real-world data, we adopt a labeling strategy similar to EmbodiedCoT [95], leveraging vision-language models to annotate object locations from visual inputs. For simulation data, we exploit the availability of camera intrinsics and extrinsics to project 3D gripper positions into 2D image coordinates. Object bounding boxes can also be directly extracted using simulator-provided segmentation masks, enabling efficient and accurate annotation of the visual scene.

Finally, we represent language planning in the following format:

  • Task: A concise natural language description of the goal the robot needs to achieve.

- Plan: A high-level sequence of steps to accomplish the task, typically numbered and described in imperative language.
- Subtask: A mid-level action derived from the plan, typically one step at a time, to be executed next.
- Subtask Reason: A rationale explaining why the current subtask is necessary or meaningful in context.
- Move: A specific low-level movement command to guide the robot toward completing the subtask.
- Move Reason: A justification of the chosen movement, often grounded in spatial alignment or task constraints.
- Gripper Position: A list of 2D coordinates that define the intended trajectory or position of the robot's gripper in image space. This often reflects the gripper's pixel-level alignment with the target object.
- Object Bounding Boxes: A list of objects currently detected in the scene, each annotated with a bounding box in pixel coordinates $[x_1, y_1, x_2, y_2]$

## B.2 Visual Reasoning Dataset

This dataset will generate visual representations in the language format for task planning. Compared to pure language-based representations, these visual representations have better spatial semantic information and are more grounded in the input images, which are used in recent multi-modal learning works [61, 86]. In this work, we use three keys, including `object bounding boxes`, `end-effector flow`, and `target object affordance` as the visual planning representations.

As shown in Figure 3, we use discrete location tokens on the input image to represent visual planning results. For an image with width $W$ and height $H$, we evenly divide both the width and height into $P$ segments each, thus we use $P \times P$ discrete bins to represent the visual pictures and each bin consists of $(W/P) \times (H/P)$ pixels. We use a new location token `<loc i>` to represent the $i$-th bin token from top-left to bottom-right, and increase the tokenizer's word vocabulary to add these bin tokens. For bounding boxes, we use the top-left and bottom-right bins to represent them. For end-effector flows, we use a sequence of bins to represent them. For affordances, we use a region of bins to represent the target regions. In this work, $W = H = 224$, and $P = 32$, i.e., each bin consists of $7 \times 7$ pixels. For example, for the task *Put the cream cheese box and the butter in the basket*, the visual reasoning data should be:

```
VISUAL OBJECT BBOXES: alphabet soup <loc_500, loc_632>, cream cheese <
loc_353, loc_452], tomato sauce <loc_461, loc_624>, ketchup <loc_341,
loc_503>, orange juice <loc_538, loc_767>, milk <loc_563, loc_791>,
butter <loc_684, loc_783>, basket <loc_448, loc_775>. VISUAL EE FLOW:
loc_387, loc_387, loc_387, loc_419, loc_419, loc_419, loc_419, loc_419
, loc_419, loc_419, loc_419, loc_451, loc_451, loc_451, loc_451,
loc_451>. VISUAL AFFORDANCE: loc_354, loc_355, loc_356, loc_386,
loc_387, loc_388, loc_418, loc_419, loc_420>
```

Specifically, given a manipulation task $\mathcal{T}$ consisting of $N$ steps ($i.e. 1, 2, ..., N$), we take the following steps to generate visual-based planning representations $\{\mathcal{V}_i^{box}, \mathcal{V}_i^{flow}, \mathcal{V}_i^{afford}\}_{i=1}^N$:

1. **Object Bounding Boxes:** We first get instance semantic maps $S = \{S_i\}_{i=1}^N \in \mathcal{R}^{N \times H \times W}$ from the simulation engine to compute binary masks for each object in each frame. Next, we sequentially apply $cv2.morphologyEx()$ to reduce noise and reconnect fragmented regions, $cv2.findContours()$ to detect object contours, and $cv2.boundingRect()$ to compute the rectangular bounding box for each detected object. Finally, we annotate the location token of the top-left and bottom-right bins for each bounding box. The final bounding box visual annotation for task $\mathcal{T}$ can be formulated as $\{\mathcal{V}_i^{box}\}_{i=1}^N$, where $\mathcal{V}_i^{box} = \{(loc_j^{tl}, loc_j^{br})\}_{j=1}^{m_i}$.

2. **End-effector Flow:** The end-effector flow visual annotation is obtained by directly labeling the location tokens corresponding to the gripper positions in the language-based planning representation. Formally, the end-effector flow annotation for task $\mathcal{T}$ can be formulated as $\{\mathcal{V}_i^{flow}\}_{i=1}^N$, where $\mathcal{V}_i^{flow} = loc_i^{gripper}$.

3. **Object Affordance:** The object affordance is represented as a heatmap centered on the target object to be fetched. We first identify the target object by detecting changes in all bounding boxes (e.g. shifts in location or variations in size). Next, we employ the pretrained SAM2 [68] model to infer a precise object mask within the target bounding box. Finally, we compute a Gaussian heatmap centered at the gripper position within the object mask to model the affordance. Location tokens corresponding to regions with affordance values exceeding a predefined threshold are then annotated in a top-left to bottom-right order. The final object affordance annotation for task $\mathcal{T}$ can be formulated as

$$\left\{\mathcal{V}_i^{afford}\right\}_{i=1}^N, \quad \text{where } \mathcal{V}_i^{afford} = \{loc_j\}_{j=1}^{n_i}.$$

## B.3 Image Foresight Reasoning

Image Foresight (IF) reasoning dataset will imagine a future goal frame as the most general representation for task planning. There is no special effort here to label the goal image. We just select the future image from the trajectory.

Here we want to introduce more about the image generation head. In this work, we use an image generation head for planning based on [31]. It auto-regressively generates the image in a coarse-to-fine paradigm proposed by [76]. Given an input image, it iteratively quantizes the residual image following a coarse-to-fine resolution schedule $\{(h_k, w_k)\}_{k=1}^K$. It also applies a technique called **B**itwise **S**elf-**C**orrection (BSC) to mitigate the performance gap between training and testing caused by teacher-forcing training.

Formally, inside each quantization iteration $k$, the tokenizer does the following steps:

1. **Calculate and Quantize Residual:** It computes the difference between the original raw feature $\boldsymbol{F}$ and the reconstructed flipped feature from the previous iteration ($\boldsymbol{F}_{k-1}^{flip}$). This residual is then interpolated to the current resolution $(h_k, w_k)$ and quantized following [90] to produce tokens at the current resolution $\boldsymbol{R}_k = \text{quantize}(\text{down}(\boldsymbol{F} - \boldsymbol{F}_{k-1}^{flip}, (h_k, w_k)))$.

2. **Apply Random Flipping For BSC:** A random flipping operation ($\text{Random\_Flip}(\cdot)$) is applied to the quantized residual $\boldsymbol{R}_k$ based on a probability $p$. This results in the flipped residual $\boldsymbol{R}_k^{flip} = \text{Random\_Flip}(\boldsymbol{R}_k, p)$.

3. **Reconstruct Flipped Feature:** The algorithm reconstructs the cumulative flipped feature $\boldsymbol{F}_k^{flip}$ up to the current iteration. It does this by interpolating *all* previously generated flipped residuals ($\boldsymbol{R}_i^{flip}$ for $i$ from 1 to $k$) to the original image resolution $(h, w)$ and sums them together: $\boldsymbol{F}_k^{flip} = \sum_{i=1}^k \text{up}(\boldsymbol{R}_i^{flip}, (h, w))$.

During inference, generation starts from a global conditioning signal, for example, the text embedding in a T2I generation setting. Notably, it generates *all* tokens of a resolution at once, distinguishing this method from the raster-scanning generation paradigm.

We select [31] as our image generation head based on three primary advantages. Firstly, it surpasses the state-of-the-art diffusion-based models [15, 21, 63] in performance on academic benchmarks and in human preference evaluations. Secondly, [31] achieves lower inference latency compared to prevalent diffusion models, a critical requirement for embodied planning within the hierarchical VLA framework. Third, our experiments indicate that the training loss of [31] serves as a stronger predictor of the final quality of foresight image generation while necessitating fewer hyperparameter adjustments, such as the noise scheduling required by the diffusion models. In practice, when the loss drops below 0.1, it indicates that the training is complete.

# C VLA-OS Model Details and Continual Learning Experiments

## C.1 Action Head Details

The action head for all VLA-OS models is in the same architecture, with only different numbers of layers. It is a block-wise causal attention transformer with the same number of layers as the LLM backbone, as introduced in Section 3.2. Let $[KV_1, \cdots, KV_n]$ be the KV tokens from the LLM, $[t]$ be the denoising timestep embedding token, $[q]$ be the proprioceptive token, and $[a_t, \cdots, a_{t+H-1}]$ be the action token, sequentially. The tokens in each block can attend to itself and blocks before it, but cannot attend to blocks after it. The hyperparameters of the action head are shown in Table 5.

Table 5: Hyperparameter of the action head transformer.

|  | Layer Number | Hidden Size | Dropout | Head | Non-Linear Func |
|---|---|---|---|---|---|
| Action Head S | 24 | 512 | 0.1 | 8 | GELU |
| Action Head S | 28 | 512 | 0.1 | 8 | GELU |
| Action Head S | 36 | 512 | 0.1 | 8 | GELU |
| Action Head S | 28 | 512 | 0.1 | 8 | GELU |

The low-level action head used for VLA-OS-H is also a transformer. It has a separate convolutional neural network (CNN) for encoding the input images, the visual planning images, and the image foresight image. For other parts, it keeps the same setting as the normal action head.

## C.2 Planning Head Details

All three planning head transformers share the same network structure (the VAE encoder and decoder of the image foresight planning head are frozen). The planning head takes as input the keys and values from each layer of the LLM backbone. The hyperparameters of the planning head are shown in Table 6.

Table 6: Hyperparameter of the planning head transformer.

|  | Layer Number | Hidden Size | Dropout | Head | Non-Linear Func |
|---|---|---|---|---|---|
| Action Head S | 24 | 512 | 0.1 | 8 | GELU |
| Action Head S | 28 | 512 | 0.1 | 8 | GELU |
| Action Head S | 36 | 512 | 0.1 | 8 | GELU |
| Action Head S | 28 | 512 | 0.1 | 8 | GELU |

## C.3 Training Loss Details

The action heads can be trained with either L1 behavior cloning loss, or the flow matching loss. L1 loss is shown to be better than L2 MSE loss for VLA [43, 44]. The L1 loss is:

$$\mathcal{L}_{L1}(\theta) = \mathbb{E}_{s,a \in \mathcal{D}} \big| \pi_\theta(s) - a \big|. \tag{1}$$

The flow matching loss is:

$$\mathcal{L}_{FM} = \mathbb{E}_{\epsilon \sim \mathcal{N}(0,I), t \sim U(0,1), s,a \sim \mathcal{D}} \big|\big| \pi_\theta(x_t, t|s) - u_t \big|\big|_2^2, \tag{2}$$

where $x_t = (1-t)\epsilon + ta$ and $u_t = \frac{d}{dt}x_t = a - \epsilon$.

The planning head losses for the language planning head and the visual planning head are the standard next-token prediction loss. The loss for the image foresight planning head follows the original paper [31].

## C.4 Training Details of Hierarchical-VLAs

The training process of Hierarchical-VLAs can have multiple choices, since they inherently incorporate two models that are not connected by backward gradients. In this work, we aim to reuse

the trained model to the greatest extent possible to reduce the cost of repeated training. Thus, for Hierarchical-VLAs, we first borrow the trained VLMs backbone as well as the planning heads from Integrated-VLAs, and finetune them on the planning datasets to get the high-level models for the Hierarchical-VLAs. Later, for the low-level model, we extract the Keys and Values from the high-level LLM backbone and use them as the embedding of the input visual and language signals and send them to each layer of the low-level action head. For the planning outputs from the high-level planning heads, we use different models to encode them: we use a frozen Qwen-2.5 7B [89] model to encode the language planning outputs to get the sentence embeddings, a common Convolutional Neural Network (with 6-channel inputs of the current 3-channel image and a 3-channel visual planning results) to encode the visual planning outputs, and a common Convolutional Neural Network (with 3-channel inputs of the goal image) to encode the image foresight planning outputs. The gradient of the low-level action head will not go backward through the high-level VLM backbone.

### C.5 Continual Learning Experiments of Different VLA Paradigms

Continual learning for robot imitation learning [51, 24] measures the degree to which the VLA model forgets old tasks when continuously learning new tasks. In this part, we test the continual learning capacities of three paradigms and three representations on 10 tasks of LIBERO-LONG sequentially. We only use Sequential Finetuning (SEQL) as our lifelong learning algorithm. We use the original metrics from LIBERO [51], including forward transfer (FWT), negative backward transfer (NBT), and area under the success rate curve (AUC). Denote $c_{i,j,e}$ as the agent's success rate on task $j$ when it learned over $i-1$ previous tasks and has just learned $e$ epochs ($e \in 0, 2, \cdots, 20$) on task $i$. Let $c_{i,i}$ be the best success rate over all evaluated epochs $e$ for the current task $i$ (i.e., $c_{i,i} = \max_e c_{i,i,e}$). Then, we find the earliest epoch $e_i^*$ in which the agent achieves the best performance on task i (i.e., $e_i^* = \arg \min_e c_{i,i,e_i} = ci, i$), and assume for all $e \geq e_i^*, c_{i,i,e} = ci, i$. Given a different task $j \neq i$, we define $c_{i,j} = ci, j, e_i^*$. Then the three metrics are defined as follows:

$$
\mathrm{FWT} = \sum_{k \in [K]} \frac{\mathrm{FWT}_k}{K}, \quad \mathrm{FWT}_k = \frac{1}{11} \sum_{e \in \{0...50\}} c_{k,k,e},
$$

$$
\mathrm{NBT} = \sum_{k \in [K]} \frac{\mathrm{NBT}_k}{K}, \quad \mathrm{NBT}_k = \frac{1}{K-k} \sum_{\tau=k+1}^{K} (c_{k,k} - c_{\tau,k}), \tag{3}
$$

$$
\mathrm{AUC} = \sum_{k \in [K]} \frac{\mathrm{AUC}_k}{K}, \quad \mathrm{AUC}_k = \frac{1}{K-k+1} \left( \mathrm{FWT}_k + \sum_{\tau=k+1}^{K} c_{\tau,k} \right).
$$

Results are shown in Table 7 and Table 8.

Table 7: Continual learning results on LIBERO-LONG of three different VLA paradigms. The VLA-OS-I and VLA-OS-H are trained with three planning representations together.

|          | FWT($\uparrow$) | NBT($\downarrow$) | AUC($\uparrow$) |
|----------|------|------|------|
| VLA-OS-A | 0.71 | **0.32** | 0.25 |
| VLA-OS-I | 0.75 | 0.43 | 0.29 |
| VLA-OS-H | **0.80** | 0.45 | **0.32** |

Table 8: Continual learning results on LIBERO-LONG of three different planning representations (Language (L), Visual (V), and Image Foresight (IF)) on VLA-OS-I.

|    | FWT($\uparrow$) | NBT($\downarrow$) | AUC($\uparrow$) |
|----|------|------|------|
| L  | 0.72 | 0.47 | 0.26 |
| V  | 0.74 | 0.40 | **0.28** |
| IF | **0.75** | **0.39** | 0.27 |

*Finding 13: VLA paradigms with task planning (Integrated-VLA and Hierarchical-VLA), compared to the non-planning paradigm (ActionOnly-VLA), achieve higher forward transfer but incur faster forgetting.*

*Finding 14: Visually grounded planning representations deliver superior forward transfer and exhibit slower forgetting relative to language-based planning representations.*

