# OpenReview forum: "VLA-OS: Structuring and Dissecting Planning Representations and Paradigms in Vision-Language-Action Models"
_NeurIPS.cc/2025/Conference — NeurIPS 2025 poster_

### Official Review · Reviewer_X7ka · 2025-06-25

**Clarity:** 3
**Significance:** 3
**Originality:** 3
**Rating:** 5
**Confidence:** 4

**Summary:**

This paper presents a systematic study of planning representations and paradigms in Vision-Language-Action (VLA) models, introducing VLA-OS, a unified architecture supporting multiple planning approaches. The authors conduct extensive experiments across rigid/deformable objects, 2D/3D modalities, and simulation/real-world environments, evaluating ActionOnly-VLA, Integrated-VLA, and Hierarchical-VLA paradigms. Key findings include: (a) Visually grounded planning representations (e.g., visual reasoning, image foresight) outperform language-based ones in task performance, training cost, and inference speed; (b) Hierarchical-VLA achieves superior task accuracy and generalization but at higher computational cost. This work provides empirical insights into VLA design choices, with implications for future research in robot manipulation.

**Questions:**

This paper systematically compares existing paradigms but does not propose new algorithms or theories. Could you clarify what fundamentally new scientific insight or methodological innovation—beyond benchmarking results—makes this work a better fit for the main conference rather than the Datasets & Benchmarks track? For example, does your analysis reveal previously unknown limitations of current paradigms that could inspire future research directions, or does VLA-OS introduce a novel architectural capability that advances the field?

**Ethical Concerns:**

["NO or VERY MINOR ethics concerns only"]

**Final Justification:**

I find this paper to be a valuable contribution to the field and support its acceptance.

**Limitations:**

Yes.

**Quality:**

3

**Strengths And Weaknesses:**

**Strengths**

1. **Comprehensiveness**: The paper introduces VLA-OS, a well-designed, modular architecture that enables standardized comparisons across diverse planning paradigms (ActionOnly, Integrated, Hierarchical) and representations (language, visual, image foresight). By controlling for variables like network architecture and training data, the study provides the first systematic, apples-to-apples evaluation of these approaches—addressing a critical gap in VLA research where prior work often conflated design choices with paradigm efficacy. The inclusion of 6 benchmarks spanning rigid/deformable objects, 2D/3D, and sim/real tasks ensures findings are robust and broadly applicable.

2. **Empirical Insights**: Through rigorous experimentation, the paper delivers practical takeaways for VLA design, such as the consistent superiority of visual over language-based planning and the trade-offs between Hierarchical-VLA’s accuracy and its computational cost.

3. **Reproducibility & Community Resources**: The release of annotated datasets and modular code ensures the work’s utility beyond the paper. By standardizing evaluation metrics and providing pretrained models, the authors enable fair comparisons for the community.

---

**Weaknesses**

1. The core weakness of this paper lies in its lack of fundamental technical innovation. While the systematic comparison of planning paradigms is thorough, the work does not introduce any novel algorithms, architectural advancements, or theoretical insights. The proposed VLA-OS framework primarily repurposes existing components without meaningful modification or improvement. The key findings—such as the superiority of visual over language-based planning—are empirically validated but align with prior intuition in robotics and VLMs. Without a new technical mechanism, the study remains an incremental benchmarking effort rather than a conceptual or algorithmic contribution. Maybe it aligns better with the Datasets & Benchmarks Track, where reproducibility and standardized evaluation are key.

2. While the paper presents extensive experiments across multiple simulation benchmarks (LIBERO, COLOSSEUM, etc.), the limited real-world validation—only three simple deformable object tasks—raises questions about the generalizability of findings to practical robotic applications.

3. While the paper thoroughly evaluates success rates across paradigms, it provides no qualitative analysis of failure cases, leaving key questions unanswered: Why does explicit planning degrade so severely when trained from scratch? Without examining specific failure modes—such as error propagation in hierarchical planning or grounding mistakes in language-based reasoning—the work misses crucial opportunities to diagnose weaknesses.

---

> ### Author Rebuttal · Authors · 2025-07-30
>
> Thank you for your efforts in reviewing our manuscript and for your insightful questions. We address each of them in turn below:
>
> *Q1: The core weakness of this paper lies in its lack of fundamental technical innovation.*
>
> A1: Thanks for your question. Although our work is primarily experimental in nature, it introduces a number of valuable innovations in network architecture, training algorithms, and empirical findings that set it apart from purely benchmark‑driven studies. Specifically:
>
> 1. **Network Architecture**: (1) We are the first to employ multiple planning representations concurrently for VLA planning and policy learning; (2) We are the first to conduct controlled experiments in which architectures remain identical while parameter counts vary; (3) We are the first to integrate a KV‑cache mechanism to extract backbone information for planning heads; (4) We are the first to introduce a VAR‑like structure for image‑generation planning in VLA; (5) We are the first to block lower‑level gradient flow in a hierarchical VLA (pi0.5 is a concurrent work).
>
> 2. **Empirical Results**: (1) We are the first to show that training VLA from scratch can outperform large‑scale, pretrained models such as OpenVLA and pi0.5‑fast; (2) We are the first to systematically compare the advantages and disadvantages of hierarchical versus integrated paradigms—evaluating success rates, training time, generalization, and continual‑learning performance; (3) We are the first to present a comprehensive comparison of different planning representations along the same metrics.
>
> From our experimental conclusions, which directly address the goal of *revealing previously unknown limitations of current paradigms that could inspire future research*, we report the following findings:
>
> 1. Image‑foresight planning outperforms visually‑grounded planning.
>
> 2. Hierarchical paradigms generalize better than integrated paradigms.
>
> 3. Scaling up model size does not necessarily improve performance on current VLA and manipulation tasks.
>
> 4. Network architecture and training algorithms remain critical; we have not yet reached the point where scaling alone suffices.
>
> 5. Hierarchical models require substantially more training (though not inference) time.
>
> 6. Planning‑based VLA underperforms end‑to‑end VLA in continual‑learning scenarios.
>
> ---
>
> *Q2: The limited real-world validation—only three simple deformable object tasks—raises questions about the generalizability of findings to practical robotic applications.*
>
> A2: Thanks for your question. We have added more real-world experiments in Appendix A.8. Please see Fig. 8. We perform the sanity check experiments in these tasks with VLA-OS-A-S and baseline models (Diffusion Policy and OpenVLA). Due to constraints on time and computational resources, we evaluated only the success rate of the ActionOnly model in order to confirm its viability on real‑world tasks. Results are as follows:
> |    | sweep   | hang   | pair   | rearrange     | put    | Average     |
> |----------|----------|----------|----------|----------|----------|----------|
> | Diffusion Policy  | 35% | 30%  | 45% | 55% | 50%  | 43% |
> | OpenVLA  | 45% | 40% | 35% | 20%| 60% | 40% |
> | VLA-OS-A-S | 55%| 50%  | 65% | 70% | 70% | 62% |
>
> The results show that the relative performance of our model versus the baseline in real‑world experiments mirrors that observed in simulation.
>
> ---
>
> *This paper provides no qualitative analysis of failure cases, leaving key questions unanswered: Why does explicit planning degrade so severely when trained from scratch?*
>
> A3: Thank you for your question. Yes, our analysis following Finding 2 was purely speculative. To rigorously test why explicit planning underperforms implicit planning, we conducted an additional experiment in which we retrained the explicit Integrated-VLA model while blocking gradient flow from the action head back into the planning head; we refer to this variant as VLA‑OS‑I‑E‑NEW. This design is ingenious because it isolates the role of both embedding type and gradient feedback:
> 1. Compared with VLA‑OS‑I-I, the only difference is that the action head receives the planning head’s information.
> 2. Compared with VLA‑OS‑I‑E, the only difference is the removal of gradient backpropagation from the action head into the planning head.
> 3. Compared with VLA-OS-‑H, the only difference is whether the action head’s input is the decoded planning representation (in VLA‑OS‑H) or the raw KV embeddings (in VLA‑OS‑I‑E‑NEW).
>
> Due to computational constraints, we evaluated only the success rates of language, visual, and image‑foresight planning representations on LIBERO-LONG. Results are as follows:
> |            | Language Planning      | Visual Planning      | Image Foresight Planning      | Average      |
> |----------|----------|----------|----------|----------|
> | VLA-OS-I-E  | 60.5  | 52.5  | 67.5 | 60.2  |
> | VLA-OS-I-E-NEW  | 64.5  | 59.5  | 68.0  | 64.0  |
> | VLA-OS-I-I  | 68.0  | 71.0  | 72.5  | 70.8  |
> | VLA-OS-H  | 63.5 | 69.0  | 71.7 | 68.1  |
>
> The resulting success‑rate ordering is: VLA‑OS‑I‑E  <  VLA‑OS‑I‑E‑NEW  <  VLA‑OS‑I‑I  $\approx$  VLA‑OS‑H
>
> These results indicate that:
>
> 1. The gradient from the action head to the planning head degrades performance.
>
> 2. Implicit planning still outperforms explicit planning.
>
> 3. Decoded planning representations yield better performance than raw KV embeddings.

---

> > ### Comment · Reviewer_X7ka · 2025-08-06
> >
> > Thank you for your response. In light of the clarifications, and given the potential value of this work to the community, I have increased my score. I also hope that the technical contributions can be more clearly emphasized in the final version of the paper.

---

### Official Review · Reviewer_gBxN · 2025-07-02

**Clarity:** 3
**Significance:** 3
**Originality:** 3
**Rating:** 4
**Confidence:** 4

**Summary:**

This paper aims to provide a detailed analysis of the numerous designs of existing vision-language-action models, comparing their pre-training objectives, model design principles, inference efficiencies, etc. The authors built a Qwen-based VLM codebase with different VLA designs, i.e., action only (predict actions directly), integrated (model predicts actions and reason on language simultaneously), and hierarchical (separate models for planning and action). The overall analysis is solid and provides several findings that serve as a summary of existing VLA paradigms.

**Questions:**

Please check the strength and limitation section.

**Ethical Concerns:**

["NO or VERY MINOR ethics concerns only"]

**Final Justification:**

The author rebuttal has addressed most of my concerns. Therefore, I'm keeping my original rating (an accept).

**Limitations:**

yes

**Paper Formatting Concerns:**

No formatting concerns

**Quality:**

2

**Strengths And Weaknesses:**

[+] I do admire the motivation of analyzing different VLA architectural designs, especially finding a thorough and systematic experiment setting to make the arguments valid.

[+] The overall experiments are solid and serve as a nice summary of recent VLA paradigms and their design choices, shedding light on how future VLA models can or should be designed.

[+] The overall codebase of this benchmarking and analysis paper could be broadly beneficial for the field of VLA.

[-] One major concern of this paper is the lack of analysis on the data recipe used for VLA, as it is mainly discussed in a setting where we train models from scratch or fine-tune models on specific tasks. As I do feel this is a critical step for the generalization capabilities of VLA models, it seems to be an important aspect.

[-] Following the previous question, the generalization experiments in this paper are not strong enough to support finding 7, as I find the 6-7% success rate a little bit too low to make a strong argument.

[-] Another major concern is on the model performance reported, where only the -S series model performance was reported. Given that the authors have already brought a series of increasing model parameters, it's still necessary to see if model size is a critical factor (especially for generalization).

[-] Another concern about this paper is that several of the findings are more like a direct read-out from the experiments, without further extension or summary. One might want to check out answers to the broad questions raised in the last few paragraphs of this paper's introduction to have a quick go-through of the findings. Paper writing can be adjusted accordingly.

---

> ### Author Rebuttal · Authors · 2025-07-30
>
> Thank you for your efforts in reviewing our manuscript and for your insightful questions. We address each of them in turn below:
>
> *Q1: Lack of analysis on the data recipe used for VLA*
>
> A: Thanks for your question. Thank you for your question. Indeed, the choice of data “recipe” is pivotal for training VLA models. We did not analyze the training datasets used in prior VLA work, as they vary widely across studies and thus resist direct standardization. To control the “data” variable in this VLA-OS study, we unified our training corpus by selecting six representative datasets that cover diverse modalities (2D vs. 3D, rigid-body vs. deformable, real-world vs. simulated, and gripper vs. dexterous-hand end-effectors), and annotated each with three planning representations that share unified structures. This controlled variable design allowed us to isolate and rigorously evaluate our proposed research questions.
>
> You may want to ask whether incorporating additional, heterogeneous datasets would affect key performance metrics. While this is indeed an intriguing question, computational constraints precluded exhaustive experimentation over all possible dataset combinations. Instead, we chose to examine one of the most critical scenarios: whether augmenting training with data that includes planning annotations but no action labels improves task success rates (see Section 4.3, Finding 6).
>
> ---
>
> *Q2: The generalization experiments in this paper are not strong enough to support finding 7*
>
> A2: Thanks for your question. We want to say that:
>
> 1. The 6%~7\% success rate generalization results are normal and common in THE COLOSSEUM benchmarks, as shown in some related works [1][2];
>
> 2. To prove that our conclusion is statistically significant, we perform the Chi‑square test for homogeneity on our generalization ability results, where it can determine whether the proportion of "success/failure" of "three different objects" is the same (i.e. whether the success rate of each group is consistent). The null hypothesis H0 is that all groups have the same success rate. We use the results of VLA-OS-A, VLA-OS-I, and VLA-OS-H from the generalization results of Table 2(a), and find that p=0.062, which shows that we can reject the null hypothesis based on a 10\% threshold. Thus, our results are statistically significant.
>
> [1] Pumacay, Wilbert, et al. "The colosseum: A benchmark for evaluating generalization for robotic manipulation." arXiv preprint arXiv:2402.08191 (2024).
>
> [2] Shridhar, Mohit, Yat Long Lo, and Stephen James. "Generative image as action models." arXiv preprint arXiv:2407.07875 (2024).
>
> ---
>
> *Q3: Given that the authors have already brought a series of increasing model parameters, it's still necessary to see if model size is a critical factor (especially for generalization).*
>
> A3: Thanks for your question. In the appendix, we report experiments on model scalability conducted on LIBERO‑LONG—please refer to Finding 12 on our anonymous website. While other questions related to model size (e.g., the impact of planning representations at different model parameter scales, generalization performance, and continual‑learning ability across models of varying sizes) are certainly of interest, computational constraints preclude us from addressing them within the rebuttal period. Concretely, as shown in Finding 12, holding all else constant, larger models require more training steps: even after 100K training steps on LIBERO‑LONG, the 7B‑parameter VLA model fails to match the performance of the 0.5B model. Executing 100K steps on our 8×A100 cluster consumes approximately 67 GPU hours, and a single generalization study would at least entail training 4 (4 model sizes) × 3 (3 paradigms) × 3 (3 planning representations) = 36 models, which is around 2400 GPU hours (=100 days). To explore how models of different scales perform on these and other questions, we plan to evolve VLA‑OS into a comprehensive open‑source platform in the future and to seek collaborations with other research groups to carry out these experiments. We hope you will show understanding and flexibility in this regard.
>
> ---
>
> *Q4: One might want to check out answers to the broad questions raised in the last few paragraphs of this paper's introduction to have a quick go-through of the findings. Paper writing can be adjusted accordingly.*
>
> A4: Thanks for your suggestion! We will revise our writing style in the camera‑ready version by condensing the conclusions and positioning them at the end of the Introduction.

---

### Official Review · Reviewer_M3Z1 · 2025-07-02

**Clarity:** 3
**Significance:** 4
**Originality:** 3
**Rating:** 5
**Confidence:** 4

**Summary:**

This paper presents VLA-OS, a unified VLA architecture suite capable of various task planning paradigms, aiming to tackle the problem in the VLA community that current approaches in VLA are mainly based on intuitive designs and lack fair and systematic comparisons. To resolve this problem, VLA-OS unifies three mainstream paradigms, including action-only VLA, Integrated VLA, and Hierarchical VLA. Also offering an interchangeable VLM backbone available from HuggingFace, various plug-and-play heads for different representations, enabling systematic and controllable experiments. The author runs exhaustive experiments on six different benchmarks to disentangle how planning representation, paradigm, and model size/pre‑training each affect task success, generalisation, training cost, and inference speed. Except for the experiment results and conclusions drawn from it, the author also provides an in-depth analysis and clear visualization of the failure cases.

**Questions:**

In the paper the author points out that policy learning is the bottleneck compared to planning. Based on the findings in this work, what does the author think might help resolving this issue?

**Ethical Concerns:**

["NO or VERY MINOR ethics concerns only"]

**Final Justification:**

It is a good paper, I recommend accept.

**Limitations:**

The scale of the experiments is kind of limited, but I understand that this requires more resources and efforts on physical experiments. If the authors can supplement this, I would like to raise the score furthermore.

**Quality:**

4

**Strengths And Weaknesses:**

**Strength:**
Through designing a fully modular backbone and interchangeable heads, VLA-OS provides a unified platform for fair comparison of various previous methods’ contribution to performance. The author provides by far the first work to compare all three mainstream VLA paradigms under identical data training and computation.

The experiment spans 2D, 3D, in-simulation, real-world scenarios, rigid-body objects, deformable objects, and different robots. Serves as a strong support to the unique findings introduced by the author.

The author also provides a new metric that separates the investigation of task planning and policy learning parts by introducing the decomposition score and instruction following scores. Provides an isolated and clear analysis of the failure cases in long-horizon tasks.
In addition to performance, the author also analyzes different paradigm’s impact on efficiency.

**Weakness:**
Besides designing a platform for unifying existing methods, it would be good for the author to integrate the findings and propose a novel architecture out of it.

Can the experiment run on larger datasets like open-X embodiment? Do the authors think the experiment results still hold?

---

> ### Author Rebuttal · Authors · 2025-07-30
>
> Thank you for your thorough review of our paper, for your endorsement of our work, and for the insightful questions you have posed. We address each of your points below in turn:
>
> *Q1: Besides designing a platform for unifying existing methods, it would be good for the author to integrate the findings and propose a novel architecture out of it.*
>
> A1: Thanks for your question. Although our work is primarily experimental in nature, it introduces a number of valuable innovations in network architecture, training algorithms, and empirical findings that set it apart from purely benchmark‑driven studies. Specifically:
>
> 1. **Network Architecture**: (1) We are the first to employ multiple planning representations concurrently for VLA planning and policy learning; (2) We are the first to conduct controlled experiments in which architectures remain identical while parameter counts vary; (3) We are the first to integrate a KV‑cache mechanism to extract backbone information for planning heads; (4) We are the first to introduce a VAR‑like structure for image‑generation planning in VLA; (5) We are the first to block lower‑level gradient flow in a hierarchical VLA (pi0.5 is a concurrent work).
>
> 2. **Empirical Results**: (1) We are the first to show that training VLA from scratch can outperform large‑scale, pretrained models such as OpenVLA and pi0.5‑fast; (2) We are the first to systematically compare the advantages and disadvantages of hierarchical versus integrated paradigms—evaluating success rates, training time, generalization, and continual‑learning performance; (3) We are the first to present a comprehensive comparison of different planning representations along the same metrics.
>
> Moreover, the VLA-OS family itself integrates the best practices from several state-of-the-art works, such as multi-step historical inputs, action-chunking outputs, and the KV-cache mechanism for extracting Transformer embeddings, and no existing VLA architecture is the same as VLA-OS models.
>
> ---
>
> *Q2: Can the experiment run on larger datasets like open-X embodiment? Do the authors think the experiment results still hold?*
>
> A2: Thank you for your suggestion! Indeed, pre‑training on a larger-scale dataset would enable us to investigate many more interesting questions. However, as you have pointed out, training such models requires substantial computational resources and time, which is an investment that is typically beyond the means of a university lab. Concretely, pre‑training a 7B‑parameter end‑to‑end VLA model with OXE consumes approximately 21,500 A100 GPU hours, which on our 8×A100 machine amounts to roughly 112 days, not to mention the hundreds of additional planning‑related ablation studies we intend to run. Consequently, for this NeurIPS submission, we do not plan to undertake large‑scale experiments.
>
> Nevertheless, we aim to build VLA‑OS into a general‑purpose VLA research open-source platform and, upon acceptance of this manuscript, to seek future collaborations with other institutions for follow‑up works such as novel research challenges that large‑scale pre‑training will bring.

---

> > ### Comment · Reviewer_M3Z1 · 2025-08-09
> > **Addressed my concerns**
> >
> > Thanks for the rebuttal. The authors addressed most of my concerns, I will maintain my score.

---

### Official Review · Reviewer_VjBW · 2025-07-03

**Clarity:** 3
**Significance:** 2
**Originality:** 2
**Rating:** 4
**Confidence:** 3

**Summary:**

The paper introduces VLA-OS, a unified architecture suite that enables controlled comparisons of three paradigms—ActionOnly, Integrated, and Hierarchical—under multiple planning representations. Systematic experiments across 2 D/3 D, rigid/deformable, simulation/real-world settings yield ten distilled findings. Key take-aways: (1) visually grounded planning beats language planning; (2) Hierarchical-VLA offers the best success and generalization but is slower to train/infer.

**Questions:**

1. Provide quantitative quality checks (e.g., 200-sample human audit) for Gemini-generated planning annotations—accuracy, inter-rater agreement, bias metrics.
2. Add the ablation where VLA-OS-I-E’s action head also consumes raw images/commands; report whether performance recovers.
3. Explicitly compare with RoboVLM and similar meta-studies—highlight what VLA-OS uncovers that predecessors missed.
4. Include and discuss the following missing citation:

Li, H., Li, M., Cheng, Z.-Q., Dong, Y., Zhou, Y., He, J.-Y., Dai, Q., Mitamura, T., & Hauptmann, A. (2025). Human-aware vision-and-language navigation: Bridging simulation to reality with dynamic human interactions. Advances in Neural Information Processing Systems, 37, 119411–119442.

**Ethical Concerns:**

["NO or VERY MINOR ethics concerns only"]

**Final Justification:**

After discussion with the author, I still believe that the novelty of the paper is missing, and some conclusions can be transferred or inferred in other similar papers. Even if the author uses more experiments to prove it.

**Limitations:**

The authors acknowledge limited coverage of latent-action and video-planning variants and the <10 k trajectory scale. Societal risks are minimal; no further concerns.

**Quality:**

3

**Strengths And Weaknesses:**

**Strengths**:
- This paper provides a unified and composable testbed to isolate and systematically study the impact of planning paradigms and representations.
- This paper introduces an original and insightful method (DCS and IFS) to separately evaluate the task planning and policy learning components of a model.
- It structures its results into ten distinct and easy-to-digest findings.


**Weaknesses**:
- **Lack of Information about Dataset Building**: The language and visual reasoning datasets were generated using Gemini, with a "Consistency Filtering" step mentioned in the pipeline. However, the paper lacks a detailed report on the quality control and validation of these crucial annotations. Without information on human verification, accuracy metrics against a ground truth, or potential biases in the automatically generated labels, the quality of the datasets remains unsubstantiated. This is a weakness because the core claims about the superiority of different planning representations rest on the correctness of this data.

- **Lack of Analysis about experiments**: The paper reports a stark performance drop for the explicit Integrated-VLA paradigm (VLA-OS-I-E) and hypothesizes that this is due to planning accumulation errors and the action head being cut off from raw sensory inputs. This is a plausible explanation, but the analysis would be stronger with more direct evidence. An ablation study that provides the action head in the VLA-OS-I-E model with raw observations—in addition to the planning embeddings—would have been a powerful way to test this hypothesis and offer a more conclusive explanation for the failure.

- **Redundant conclusion**：Some findings are similar to RoboVLM [1], such as the performance differences in training from scratch, of course, the author distinguishes more architectures and situations.

---

> ### Author Rebuttal · Authors · 2025-07-30
>
> Thank you for your efforts in reviewing our paper. We appreciate your insightful summary and the questions you have raised. We address each of your points below:
>
> *Q1: Lack of Information about Dataset Building*
>
> A1: We thank the reviewer for highlighting the importance of dataset quality. As noted in Appendix B.1, **consistency filtering** addresses two issues: (1) incorrect planning and (2) inconsistent planning across trajectories of the same task.
>
> For (1), we report the initial reasoning accuracy based on human verification. For example, in LIBERO-LONG (379 samples), the initial reasoning accuracy is 0.71 (269/379), where most Gemini-generated plans are correct, and a high-quality template can be selected automatically via Gemini. In contrast, in harder datasets like FurnitureBench, nearly all initial plans are incorrect due to long-horizon complexity, and the initial reasoning accuracy can be regarded as 0. In such cases, we manually create the planning template to ensure regeneration works as intended.
>
> For (2), although consistency is hard to quantify, our pipeline guarantees that all annotations for the same task are identical after filtering, as Gemini follows a fixed high-quality prompt. The high quality can be verifiable in the released dataset.
>
> ---
>
> *Q2: Lack of Analysis about the explicit planning v.s. implicit planning experiments*
> A2: Thank you for your question. Yes, our analysis following Finding 2 was purely speculative. To rigorously test why explicit planning underperforms implicit planning, we conducted an additional experiment in which we retrained the explicit Integrated-VLA model while blocking gradient flow from the action head back into the planning head; we refer to this variant as VLA‑OS‑I‑E‑NEW. This design is ingenious because it isolates the role of both embedding type and gradient feedback:
> 1. Compared with VLA‑OS‑I-I, the only difference is that the action head receives the planning head’s information.
> 2. Compared with VLA‑OS‑I‑E, the only difference is the removal of gradient backpropagation from the action head into the planning head.
> 3. Compared with VLA-OS-‑H, the only difference is whether the action head’s input is the decoded planning representation (in VLA‑OS‑H) or the raw KV embeddings (in VLA‑OS‑I‑E‑NEW).
>
> Due to computational constraints, we evaluated only the success rates of language, visual, and image‑foresight planning representations on LIBERO-LONG. Results are as follows:
> |            | Language Planning      | Visual Planning      | Image Foresight Planning      | Average      |
> |----------|----------|----------|----------|----------|
> | VLA-OS-I-E  | 60.5  | 52.5  | 67.5 | 60.2  |
> | VLA-OS-I-E-NEW  | 64.5  | 59.5  | 68.0  | 64.0  |
> | VLA-OS-I-I  | 68.0  | 71.0  | 72.5  | 70.8  |
> | VLA-OS-H  | 63.5 | 69.0  | 71.7 | 68.1  |
>
> The resulting success‑rate ordering is: VLA‑OS‑I‑E  <  VLA‑OS‑I‑E‑NEW  <  VLA‑OS‑I‑I  $\approx$  VLA‑OS‑H
>
> These results indicate that:
>
> 1. The gradient from the action head to the planning head degrades performance.
>
> 2. Implicit planning still outperforms explicit planning.
>
> 3. Decoded planning representations yield better performance than raw KV embeddings.
>
> ---
>
> *Q3: Comparison with RoboVLMs*
>
> A3: Thank you for raising this point. While RoboVLMs also offers a systematic investigation of VLA models (and is duly cited in our Related Work Section), VLA‑OS differs fundamentally in both research objectives and research scope:
>
> 1. **Difference in architectural emphasis**. VLA‑OS employs a single, unified network architecture to study how to leverage VLA—deliberately fixing the backbone to de‑emphasize architectural design. By contrast, RoboVLMs addresses the lower-level question of how to construct VLA networks, systematically evaluating different VLM backbones, input‑history lengths, action‑sequence lengths, and discrete vs. continuous action spaces. Rather than re‑exploring network design, VLA‑OS builds upon those findings, integrating the most effective components (and complementary insights from other work such as $\pi_0$) into the VLA‑OS model family.
>
> 2. **Difference in research scope**. VLA‑OS is centered on task planning and the interplay between planning and policy learning, exploring issues such as planning data construction, representation selection, and comparative analysis of planning paradigms. In contrast, RoboVLMs focuses on purely end‑to‑end (planning‑free) VLA models versus non‑VLA baselines, evaluating primarily whether VLA yields superior performance and generalization (for example, compared to simple diffusion policies).
>
> ---
>
> *Q4: Comparison with Human-aware vision-and-language navigation: Bridging simulation to reality with dynamic human interactions*
>
> A4: Thank you for bringing this outstanding related work to my attention. We will incorporate this paper into our references in the camera-ready version.
>
> This work presents a significant advance in embodied AI by integrating dynamic human activities into traditional VLN benchmarks, thereby narrowing the sim‑to‑real gap through three key innovations: an egocentric 60° action space, a realistic HA3D simulator populated with SMPL‑based human motion models, and the HA‑R2R dataset that enriches navigation instructions with human activity contexts. This work complements the VLA‑OS studies in robotic manipulation by similarly emphasizing the separation of high‑level planning and low‑level execution through unified architectures and large‑scale data, while extending VLA paradigms to the domain of dynamic, human‑populated environments.

---

### Decision · Program_Chairs · 2025-09-17

**Decision:**

Accept (poster)

**Comment:**

This paper introduces VLA-OS, a unified framework for systematically comparing Vision-Language-Action (VLA) models across major paradigms (ActionOnly, Integrated, Hierarchical) and planning representations. Through extensive experiments in diverse simulation and real-world settings, the study provides clear empirical insights: visually grounded planning generally outperforms language-based planning, and hierarchical approaches achieve the best accuracy and generalization at higher computational cost.

**Strengths**:
- Comprehensive framework: Provides systematic comparison across paradigms and planning representations, filling a gap in the field.
- Empirical insights: Yields nontrivial findings (e.g., implicit planning outperforming explicit, training-from-scratch surpassing pretrained VLAs, hierarchical methods offering better generalization).
- Community impact: Open-sourcing models and benchmarks enhances reproducibility and utility.
- Experimental rigor: Large-scale evaluation across diverse simulated and real-world tasks.

**Weaknesses**
- Limited technical novelty: Framework reuses existing components; architectural innovations are modest and may be viewed as incremental.
- Real-world validation: Although expanded in the rebuttal, real-world experiments remain relatively small-scale compared to the breadth of simulation.
- Qualitative analysis: Initial submission lacked deep error analysis; although partly addressed in rebuttal, broader interpretation and insights could still strengthen the contribution.

**Rebuttal & Discussion**
Reviewer concerns centered on technical novelty, real-world generalization, and lack of diagnostic analysis. Authors responded with clarifications of architectural/algorithmic contributions (concurrent planning, VAR-like foresight, gradient-blocking in hierarchical planning). The novelty concern remains partly unresolved; while the authors’ clarifications highlight thoughtful design, many reviewers may still see the work as primarily empirical.

**Suggestions for Improvement**
- Expand the real-world validation to cover more diverse and complex tasks.
- Strengthen the presentation of technical contributions, making clear what is novel beyond systematic benchmarking.
- Include deeper qualitative analysis and discussion of failure modes and design trade-offs.

This is a strong empirical contribution with clear community value: a unified framework, open-source release, and careful benchmarking that yields actionable insights for future VLA research. While the level of technical innovation is debatable, the thoroughness, rigor, and clarity of results justify acceptance.